# Client-Private Secure Aggregation for Privacy-Preserving Federated Learning

**Parker Newton**
Amazon Web Services
pnnewto@amazon.com

**Olivia Choudhury**
Amazon Web Services
olichou@amazon.com

**Bill Horne**
Amazon Web Services
bgh@amazon.com

**Vidya Ravipati**
Amazon Web Services
ravividy@amazon.com

**Divya Bhargavi**
Amazon Web Services
dbharga@amazon.com

**Ujjwal Ratan**
Amazon Web Services
ujjwalr@amazon.com

## Abstract

Privacy-preserving federated learning (PPFL) is a paradigm of distributed privacy-preserving machine learning training in which a set of clients, each holding siloed training data, jointly compute a shared global model under the orchestration of an aggregation server. The system has the property that no party learns any information about any client's training data, besides what could be inferred from the global model. The core cryptographic component of a PPFL scheme is the secure aggregation protocol, a secure multi-party computation protocol in which the server securely aggregates the clients' locally trained models into an aggregated global model, which it distributes to the clients. However, in many applications the global model represents a trade secret of the consortium of clients, which they may not wish to reveal in the clear to the server. In this work, we propose a novel model of secure aggregation, called client-private secure aggregation (CPSA), in which the server computes an encrypted global model which only the clients can decrypt. We provide three explicit constructions of CPSA which exhibit varying trade-offs. We also conduct experimental results to demonstrate the practicality of our constructions in the cross-silo setting when scaled to 250 clients.

## 1 Introduction

*Federated learning* (FL) [24] is a paradigm of distributed machine learning (ML) training in which $n$ clients, each holding siloed training data, jointly compute a shared global model under the orchestration of an aggregation server, without revealing their training data in the clear. This is particularly applicable in industries where sensitive data is distributed across silos and centralizing such data for analysis is infeasible. For example, in the healthcare domain, a consortium of healthcare providers, each hosting patient-level sensitive data, can contribute towards building a shared machine learning model to improve patient care, while aiding in complying with regulatory guidelines [10, 11, 12]. Also, in the financial services industry, FL has been applied to leverage data hosted across a consortium of banks to improve their joint ability to detect credit card fraud [32]. In FL, the aggregation server maintains the current state of the global model so that when new clients join, the global model can be distributed to the new clients. Additionally, the aggregation server facilitates communication between the consortium of clients so that the clients need not establish a complete network graph to directly communicate with each other. In this work, we consider cross-silo FL, in which the clients are typically fixed institutions (e.g. banks, hospitals, research institutions, etc.), and total in number on the order of 100.

Workshop on Federated Learning: Recent Advances and New Challenges, in Conjunction with NeurIPS 2022 (FL-NeurIPS'22). This workshop does not have official proceedings and this paper is non-archival.

FL begins with the server sending the initial global model to all clients. Each client then locally trains the model on their training data to compute a local model update, which they send to the aggregation server. The server aggregates the model updates from the clients to compute a global model update, which it applies to the initial model to compute a new global model. The new global model is then broadcast to the clients. This process can be repeated until a global optimum is reached.

Note that after an iteration of FL, each party receives the new aggregated global model without any client sharing their training data in the clear. However, several attacks [31, 33] demonstrate that an adversary can partially reconstruct the clients' training data from their local model updates. The breakthrough work of [3] constructed a secure multi-party computation (MPC) protocol in which the server computes the sum of the clients' local model updates, which it then broadcasts to the clients. The security of the protocol enforces that no party learns any information about any client's model update, except for what could be inferred from the sum of the clients' model updates. FL in which the plaintext aggregation of clients' local model updates in replaced with a secure aggregation protocol is called *privacy-preserving federated learning* (PPFL). In PPFL, instead of each client supplying their local model update as input, each client can simply supply their local model weights.

In the standard model of secure aggregation, the server ultimately computes the global model in the clear, which it then broadcasts to the clients. However, in many cases the global model represents a trade secret of the consortium of clients, which they may not wish to leak to the server. For example, consider the case in which a consortium of pharmaceutical companies wish to use PPFL vended by a cloud service provider (CSP) to aid with drug discovery. In this case, the global model is a trade secret of the consortium, which they do not wish to leak to the CSP.

In this work, we propose a novel model of secure aggregation, called *client-private secure aggregation* (CPSA), in which the server computes an encrypted global model which only the clients can decrypt. CPSA still enjoys the security of the plain model of secure aggregation (that is, no adversarially corrupted subset of parties learns any information, beyond the protocol output, about any non-corrupted client's input), with the additional security guarantee that an adversarial server learns no information about any client's input, not even the protocol output. CPSA can be combined with differential privacy (DP) to additionally enforce that any information which an adversarially corrupted subset of clients can infer from the plaintext protocol output about *some* non-corrupted client's input cannot be associated with any particular client. Finally, since in the cross-silo FL setting, the network availability of the clients is typically not an issue, we don't seek to address client dropout in our CPSA model.

**Prior Work.** Beginning with the foundational work of [3], secure aggregation protocols have been widely studied in the literature [2, 6, 7, 23, 29]. Various protocols have been constructed which offer trade-offs with respect to the security model and computational, communication, space, and round complexity. The original work of [3] constructed a four-round secure aggregation protocol with semi-honest security (and a five-round variant with malicious security). Their protocol works by each client choosing a random mask which locally encrypts their input as a one-time pad (OTP), but with the property that the clients' masks all together sum to zero. In this way, the server can sum over the OTP's from the clients to compute the sum of their inputs.

In [29], the authors construct a natural two-round computationally efficient secure aggregation protocol with semi-honest security using threshold additive homomorphic encryption (AHE). The work of [2] employs a simple additive secret sharing approach to achieve a one-round maliciously secure aggregation protocol. Their protocol is computationally and communication-efficient, but requires two independent non-colluding servers.

Recall that the security property of a secure aggregation protocol enforces that no party learns any information about any other party's input, except for what could be inferred from the protocol output. This begs the question if we can enforce a privacy guarantee against an adversary inferring information about a party's input from the protocol output. Differential privacy (DP) [14, 15] is a statistical model of privately releasing aggregate data which masks a single party's contribution to the aggregate data. That is, DP ensures that no adversary, given access to the differentially private aggregate data, can infer any information about any particular party's contribution to the aggregate data. Several secure aggregation protocols [6, 7, 20] employ DP to construct a protocol in which all parties compute a differentially private sum of the clients' inputs.

**Our Contributions.** We introduce a novel model of secure aggregation for PPFL, called *client-private secure aggregation* (CPSA), in which the server computes an encrypted global model which only the clients can decrypt. CPSA protocols combined with differential privacy achieve complete input privacy for the clients during PPFL, precluding model inversion and membership inference attacks against all parties. Additionally, this model enforces a stronger security guarantee against the server, namely that an adversarial server learns no information about any client's input, not even the protocol output. We construct three novel CPSA protocols, $\Pi_0, \Pi_1, \Pi_2$, which are secure against a semi-honest adversary, and each offer varying trade-offs. If $m \in \mathbb{N}$ is the dimension of each client's input vector to the protocol, then we define $m' = m'(m, n) \in \mathbb{N}$ as the dimension of the server's output ciphertext vector. $\Pi_0$ is a two-round protocol, while the other two both require three rounds. $\Pi_1$ and $\Pi_2$ both achieve a constant-rate output ciphertext vector dimension of $m' = m$, while $\Pi_0$ has $m' = \mathcal{O}(mn^2)$. $\Pi_1$ is more communication efficient over $\Pi_0$ and $\Pi_2$, but the decryption key used by the clients in $\Pi_1$ to decrypt the ciphertext output by the server is not reusable over multiple iterations of CPSA for PPFL. Consequently, when new clients join the collaboration, the trusted setup algorithm which distributes the decryption key to the new clients must interact with the protocol participants to update the decryption key after each iteration. On the other hand, $\Pi_2$ supports reusable decryption keys, and so the trusted setup algorithm need not function as an active participant in the protocol. Finally, we provide an empirical evaluation of our protocols to demonstrate their practicality when scaled to 250 clients. We remark that we limit the scope of this work to studying the CPSA protocol itself. A plethora of prior works ([2, 6, 7, 29], and many more) have shown how to use PPFL with different secure aggregation protocols and DP to train high-quality models on popular standard datasets, and so it's clear that CPSA protocols can be similarly applied in this manner.

## 2 Preliminaries

### 2.1 Notation

If $k \in \mathbb{N}$, then we denote by $[k]$ the set $\{1, 2, \ldots, k\}$. If $q \in \mathbb{N}$, then we write $\mathbb{Z}_q$ for the ring of integers $(\bmod\ q)$. For $m \in \mathbb{N}$, we denote vectors in $\mathbb{Z}_q^m$ by bold lower-case characters $\mathbf{x}$. If $\mathbf{x} \in \mathbb{Z}_q^m$, then we denote the $i^{\text{th}}$ component of $\mathbf{x}$ by $x_i \in \mathbb{Z}_q$. If $x_1, \ldots, x_m \in \mathbb{Z}_q$, then we write $(x_i)_{i \in [m]}$ for the vector in $\mathbb{Z}_q^m$ whose $i^{\text{th}}$ component is $x_i$. Sets are written as $S$, algorithms are written as $\mathcal{A}$, and probabilistic distributions are written as $\mathsf{D}$. Throughout this work, we denote the security parameter by $\lambda \in \mathbb{N}$. A quantity $f(\lambda)$ is said to be negligible in $\lambda$, written $f(\lambda) = \mathrm{negl}(\lambda)$, if $f(\lambda)$ asymptotically tends to zero faster than any inverse polynomial in $\lambda$. A quantity $f(\lambda)$ is said to be polynomial in $\lambda$ if $f(\lambda) = \mathcal{O}(\lambda^c)$, for some constant $c \in \mathbb{N}$. We say that two distributions $\mathsf{X}$ and $\mathsf{Y}$ are statistically indistinguishable, written $\mathsf{X} \equiv \mathsf{Y}$, if for every probabilistic algorithm $\mathcal{A}$ which gives output in $\{0, 1\}$, it holds that

$$\left| \Pr_{x \sim \mathsf{X}} \left[ \mathcal{A}(1^\lambda, x) = 1 \right] - \Pr_{y \sim \mathsf{Y}} \left[ \mathcal{A}(1^\lambda, y) = 1 \right] \right| = \mathrm{negl}(\lambda). \tag{1}$$

In the aforementioned definition, if we instead restrict $\mathcal{A}$ to be a probabilistic polynomial time (PPT) algorithm, then we say that $\mathsf{X}$ and $\mathsf{Y}$ are computationally indistinguishable, and write $\mathsf{X} \approx_c \mathsf{Y}$.

### 2.2 Key Agreement Scheme

Here, we define a key agreement scheme [8], which is typically used for two parties to agree on a shared key for a symmetric-key cryptosystem.

**Definition 1.** *A key agreement scheme is a pair of PPT algorithms* $\mathsf{KA} = (\mathsf{Gen}, \mathsf{Agree})$ *with the following syntax, correctness, and security.*

**Syntax:**

- $\mathsf{Gen}(1^\lambda)$ *takes as input the security parameter* $\lambda$ *and outputs a public/secret key pair* $(\mathrm{pk}, \mathrm{sk})$ *for some user.*

- $\mathsf{Agree}(\mathrm{sk}_i, \mathrm{pk}_j)$ *takes as input a secret key* $\mathrm{sk}_i$ *corresponding to some user* $i$, *and a public key* $\mathrm{pk}_j$, *corresponding to some user* $j \neq i$, *and outputs a key* $\mathrm{k}_{i,j}$ *from the key space* $\mathsf{K}$.

**Correctness:** *Let $\lambda$ be the security parameter. If $(\mathrm{pk}_1, \mathrm{sk}_1), (\mathrm{pk}_2, \mathrm{sk}_2) \leftarrow \mathsf{Gen}(1^\lambda), \mathrm{k}_{1,2} = \mathsf{Agree}(\mathrm{sk}_1, \mathrm{pk}_2), \mathrm{k}_{2,1} = \mathsf{Agree}(\mathrm{sk}_2, \mathrm{pk}_1)$, then $\mathrm{k}_{1,2} = \mathrm{k}_{2,1}$.*

**Security:** *Let $\lambda$ be the security parameter. Define the following distributions:*

- *$\mathsf{D}_0(1^\lambda)$ : Compute $(\mathrm{pk}_1, \mathrm{sk}_1), (\mathrm{pk}_2, \mathrm{sk}_2) \leftarrow \mathsf{Gen}(1^\lambda)$, $\mathrm{k} = \mathsf{Agree}(\mathrm{sk}_1, \mathrm{pk}_2)$, and output $(\mathrm{pk}_1, \mathrm{pk}_2, \mathrm{k})$.*

- *$\mathsf{D}_1(1^\lambda)$ : Compute $(\mathrm{pk}_1, \mathrm{sk}_1), (\mathrm{pk}_2, \mathrm{sk}_2) \leftarrow \mathsf{Gen}(1^\lambda)$, $\mathrm{k} \leftarrow \mathrm{K}$, and output $(\mathrm{pk}_1, \mathrm{pk}_2, \mathrm{k})$.*

*If $\mathcal{A}$ is a PPT distinguishing algorithm, then $\forall b \in \{0, 1\}$, define*

$$\mathrm{P}_b^{\mathcal{A}}(\lambda) := \Pr_{(\mathrm{pk}_1, \mathrm{pk}_2, \mathrm{k}) \leftarrow \mathsf{D}_b(1^\lambda)} \left[ \mathcal{A}(1^\lambda, \mathrm{pk}_1, \mathrm{pk}_2, \mathrm{k}) = 1 \right]. \tag{2}$$

*Then, for all PPT distinguishing adversaries $\mathcal{A}$,*

$$\left| P_0^{\mathcal{A}}(\lambda) - P_1^{\mathcal{A}}(\lambda) \right| = \mathrm{negl}(\lambda). \tag{3}$$

## 2.3 Authenticated Encryption

Authenticated encryption (AE) is a cryptographic primitive that provides confidentiality and integrity of messages exchanged between two parties which each hold a shared symmetric key.

**Definition 2.** *An authenticated encryption scheme is a triple of PPT algorithms $\mathsf{AE} = (\mathsf{Gen}, \mathsf{Enc}, \mathsf{Dec})$ with the following syntax, correctness, and security.*

**Syntax:**

- *$\mathsf{Gen}(1^\lambda)$ takes as input the security parameter $\lambda$ and outputs a symmetric key $\mathrm{k} \in \mathrm{K}$ in the key space $\mathrm{K}$.*

- *$\mathsf{Enc}(\mathrm{k}, m)$ takes as input a symmetric key $\mathrm{k} \in \mathrm{K}$, a message $m \in \mathrm{M}$ in the message space $\mathrm{M}$, and outputs a ciphertext $c = (c', t) \in \mathrm{C}$ of $m$ under $\mathrm{k}$ in ciphertext space $\mathrm{C}$. $c'$ denotes the actual ciphertext of the message, while $t$ denotes the message authentication code (MAC).*

- *$\mathsf{Dec}(\mathrm{k}, c)$ takes as input a symmetric key $\mathrm{k} \in \mathrm{K}$ and a ciphertext $c = (c', t) \in \mathrm{C}$, and outputs either a message $m \in \mathrm{M}$ or an error symbol $\perp$.*

**Correctness:** *Let $\lambda$ be the security parameter. If $\mathrm{k} \leftarrow \mathsf{Gen}(1^\lambda)$, $m \in \mathrm{M}$ is a message, $c \leftarrow \mathsf{Enc}(\mathrm{k}, m)$, then $\mathsf{Dec}(\mathrm{k}, c) = m$.*

**Semantic Security:** *Let $\lambda$ be the security parameter. If $\mathrm{k} \leftarrow \mathsf{Gen}(1^\lambda)$, then for every PPT distinguishing adversary $\mathcal{A}$ and distinct messages $m_0, m_1 \in \mathrm{M}$, it holds that*

$$\left| \Pr_{c \leftarrow \mathsf{Enc}(\mathrm{k}, m_0)} \left[ \mathcal{A}(1^\lambda, m_0, m_1, c) = 1 \right] - \Pr_{c \leftarrow \mathsf{Enc}(\mathrm{k}, m_1)} \left[ \mathcal{A}(1^\lambda, m_0, m_1, c) = 1 \right] \right| = \mathrm{negl}(\lambda). \tag{4}$$

**Ciphertext Integrity:** *Let $\lambda$ be the security parameter. The AE scheme $\mathsf{AE} = (\mathsf{Gen}, \mathsf{Enc}, \mathsf{Dec})$ is said to provide* ciphertext integrity *if every PPT adversary $\mathcal{A}$ can only win the following game against a computationally unbounded challenger $\mathcal{C}$ with probability $\mathrm{negl}(\lambda)$:*
Setup: $\mathcal{C}$ computes $\mathrm{k} \leftarrow \mathsf{Gen}(1^\lambda)$.
Query Phase: For all $i = 1, \ldots, r = \mathrm{poly}(\lambda)$, $\mathcal{A}$ generates a message $m_i \in \mathrm{M}$ and send $m_i$ to $\mathcal{C}$. $\mathcal{C}$ then computes and outputs to $\mathcal{A}$ the ciphertext $c_i \leftarrow \mathsf{Enc}(\mathrm{k}, m_i)$.
Challenge Phase: $\mathcal{A}$ produces and sends to $\mathcal{C}$ a ciphertext $c' \in \mathrm{C}$. $\mathcal{A}$ wins if $c' \notin \{c_1, \ldots, c_r\}$ and $\mathsf{Dec}(\mathrm{k}, c') \neq \perp$.

## 2.4 Pseudorandom Generator

**Definition 3.** *Let $r, s \in \mathbb{N}$ such that $r < s$. A pseudorandom generator (PRG) [21, 22] is a PPT function $G : \{0,1\}^r \to \{0,1\}^s$ such that $G(\mathsf{U}(\{0,1\}^r)) \approx_c \mathsf{U}(\{0,1\}^s)$, where $\mathsf{U}(\{0,1\}^r)$ and $\mathsf{U}(\{0,1\}^s)$ denote the uniform distributions on $\{0,1\}^r$ and $\{0,1\}^s$, respectively.*

A PRG $G$ can be used to strecth a random shared symmetric key in the following way. Let $\mathrm{K}$ be a symmetric key space and $q, m \in \mathbb{N}$ such that $\log_2(|\mathrm{K}|) < m \log_2(q)$. Then, it is easy to see that without loss of generality we can define a PRG $G : \mathrm{K} \to \mathbb{Z}_q^m$.

## 2.5 Additive Secret Sharing

Let $n, t, q \in \mathbb{N}$. A $(t, n)-$ *secret sharing* scheme over $\mathbb{Z}_q$ is a pair of PPT algorithms (Share, Rec) with the following properties:

- Share$(x)$ takes as input a secret $x \in \mathbb{Z}_q$ and outputs shares $\{s_i\}_{i \in [n]}$ for a set of $n$ users, indexed by $[n]$.
- Rec$(\{x_{i_j}\}_{j \in [t]})$ takes as input a subset $\{x_{i_j}\}_{j \in [t]} \subseteq \mathbb{Z}_q$ of $t$ distinct shares of a secret $x \in \mathbb{Z}_q$, and reconstructs and outputs $x \in \mathbb{Z}_q$.
- Any subset of shares of size less than $t$ is statistically independent of the underlying secret.

*Additive secret sharing* is a $(n, n)-$secret sharing scheme over $\mathbb{Z}_q$ in which Share$(x)$ chooses random $s_1, \ldots, s_{n-1} \leftarrow \mathbb{Z}_q$, computes $s_n = x - \sum_{i \in [n-1]} s_i \in \mathbb{Z}_q$, and outputs $\{s_i\}_{i \in [n]}$. Rec$(\{s_i\}_{i \in [n]})$ simply works by outputting $\sum_{i=1}^{n} s_i \in \mathbb{Z}_q$. Note that any subset of $\{s_i\}_{i=1}^{n}$ of size $k < n$ is distributed identically to $k$ uniformly random elements of $\mathbb{Z}_q$, hence is statistically independent of the secret $x$.

## 2.6 Homomorphic Encryption

Homomorphic encryption (HE) [4, 5, 16, 18, 19, 26, 27] is a cryptographic primitive which enables computation directly on encrypted data. That is, HE is an encryption scheme which supports homomorphic addition or multiplication operations, so that a party, holding only ciphertexts of two messages $m_1, m_2$, can apply the homomorphic addition (resp., multiplication) operation to compute a ciphertext of $m_1 + m_2$ (resp., $m_1 \cdot m_2$). Since an arbitrary computable function $f : \{0, 1\}^* \to \{0, 1\}^*$ can be expressed as an arithmetic circuit, then theoretically a HE scheme allows a client $C$, which holds a private input $\mathbf{x} \in \{0, 1\}^*$, to outsource the computation of $f(\mathbf{x})$ to a server $S$ without revealing any information about $\mathbf{x}$ to $S$. This works by $C$ encrypting $\mathbf{x} \in \{0, 1\}^*$ and sending the ciphertext to $S$, which can homomorphically compute a ciphertext of $f(\mathbf{x})$ which can be decrypted by $C$. The semantic security of the HE scheme ensures that $S$ learns no information about $\mathbf{x}$ during the homomorphic evaluation of $f(\mathbf{x})$.

A partially homomorphic encryption (PHE) scheme is an HE scheme that supports either homomorphic addition or multiplication operations, but not both. PHE schemes are either additive homomorphic encryption (AHE) schemes or multiplicative homomorphic encryption (MHE) schemes. An example of an AHE scheme is Paillier Encryption [26], while examples of MHE schemes are RSA [28] and ElGamal Encryption [17].

A fully homomorphic encryption (FHE) scheme is a HE scheme that supports both homomorphic addition and multiplication operations. First constructed by Craig Gentry in [18], numerous follow-up works [4, 5, 16, 19] introduced improved constructions of FHE schemes. Most FHE constructions rely on a computationally expensive bootstrapping operation [1, 9, 13] to refresh the ciphertexts after a fixed-length consecutive sequence of homomorphic operations. Indeed, these bootstrapping algorithms continue to serve as the principal bottleneck in achieving FHE as a computationally practical general-purpose solution to privacy-preserving cloud-outsourced computation.

In this work, we use AHE, and so for completeness we provide a formal definition of AHE below.

**Definition 4.** *An additive homomorphic encryption (AHE) scheme is a quadruple of PPT algorithms* AHE $=$ (Gen, Enc, Dec, Add) *with the following syntax, correctness, and security.*

**Syntax:**

- Gen$(1^\lambda)$ *takes as input the security parameter* $\lambda \in \mathbb{N}$ *and outputs a public/secret key pair* $(\mathrm{pk}, \mathrm{sk})$.
- Enc$(\mathrm{pk}, m)$ *takes as input a public key* $\mathrm{pk}$ *and message* $m \in \mathrm{M}$ *in the message space* $\mathrm{M}$, *and outputs a ciphertext* $c \in \mathrm{C}$ *in the ciphertext space* $\mathrm{C}$.
- Dec$(\mathrm{sk}, c)$ *takes as input a secret key* $\mathrm{sk}$ *and ciphertext* $c \in \mathrm{C}$, *and outputs a message* $m \in \mathrm{M}$.
- Add$(c_1, c_2)$ *takes as input two ciphertexts* $c_1, c_2 \in \mathrm{C}$ *and outputs a ciphertext* $c_3 \in \mathrm{C}$.

**Correctness of Decryption:**   *Let $\lambda \in \mathbb{N}$ be the security parameter, $m \in \mathrm{M}$ be a message, and suppose $(\mathrm{pk}, \mathrm{sk}) \leftarrow \mathsf{Gen}(1^\lambda), c \leftarrow \mathsf{Enc}(\mathrm{pk}, m)$. Then, $\mathsf{Dec}(\mathrm{sk}, c) = m$.*

**Correctness of Homomorphic Addition:**   *Let $\lambda \in \mathbb{N}$ be the security parameter, $m_1, m_2 \in \mathrm{M}$ be a message, and suppose $(\mathrm{pk}, \mathrm{sk}) \leftarrow \mathsf{Gen}(1^\lambda), c_i \leftarrow \mathsf{Enc}(\mathrm{pk}, m_i) \ \forall i \in \{1, 2\}$, and $c_3 \leftarrow \mathsf{Add}(c_1, c_2)$. Then, $\mathsf{Dec}(\mathrm{sk}, c_3) = m_1 + m_2$.*

**Semantic Security:**   *Let $\lambda \in \mathbb{N}$ be the security parameter, and suppose $(\mathrm{pk}, \mathrm{sk}) \leftarrow \mathsf{Gen}(1^\lambda)$. If $\mathcal{A}$ is a PPT distinguishing algorithm and $m_0, m_1 \in \mathrm{M}$ are distinct messages, then $\forall b \in \{0, 1\}$ define*

$$\mathrm{P}_b^{\mathcal{A}}(\lambda, \mathrm{pk}, m_0, m_1) := \Pr_{c \leftarrow \mathsf{Enc}(\mathrm{pk}, m_b)}\left[\mathcal{A}(1^\lambda, \mathrm{pk}, m_0, m_1, c) = 1\right]. \tag{5}$$

*Then, for every PPT distinguishing adversary $\mathcal{A}$ and distinct messages $m_0, m_1 \in \mathrm{M}$, it holds that*

$$\left|\mathrm{P}_0^{\mathcal{A}}(\lambda, \mathrm{pk}, m_0, m_1) - \mathrm{P}_1^{\mathcal{A}}(\lambda, \mathrm{pk}, m_0, m_1)\right| = \mathrm{negl}(\lambda). \tag{6}$$

## 2.7   Differential Privacy

Differential privacy (DP) [14, 15] is a statistical model of private aggregate data release which masks a single party's contribution to the aggregate data. That is, DP ensures that no adversary, given access to the differentially private aggregate data, can infer any information about any particular party's contribution to the aggregate data. While an adversary may infer be able to infer information about some client's contribution to the aggregate data, they can't associate that inference with a particular client.

Let $X, Y \subseteq \mathbb{R}$, $n, m \in \mathbb{N}$, and $f : X^n \to Y^m$ be a function. $f$ is meant to model an aggregate function of data collected from $n$ users which is represented as a database record $\mathbf{x} \in X^n$. We define the function $\mathrm{dist}(\cdot, \cdot) : X^n \times X^n \to \mathbb{Z}$ as the hamming distance function (*i.e.,* $\mathrm{dist}(\mathbf{x}, \mathbf{x}') = \#\{i \in [n] : x_i \neq x_i'\}$). A differentially private mechanism for $f$ is a PPT algorithm $\mathcal{M}^f$ which gets oracle access to $f$, takes input in $X^n$, and provides output in $Y^m$. We now provide a formal definition of a differentially private mechanism below.

**Definition 5.** *Let $\varepsilon, \delta > 0$. A PPT algorithm $\mathcal{M}^f$ is said to be an $(\varepsilon, \delta)-$differentially private mechanism for $f$ if $\forall \mathbf{x}, \mathbf{x}' \in X^n$ such that $\mathrm{dist}(\mathbf{x}, \mathbf{x}') = 1$, and $\forall S \subseteq \mathrm{supp}(\mathcal{M})$,*

$$\Pr_{\mathbf{y} \leftarrow \mathcal{M}^f(\mathbf{x})}\left[\mathbf{y} \in S\right] \leq e^\varepsilon \cdot \Pr_{\mathbf{y}' \leftarrow \mathcal{M}^f(\mathbf{x}')}\left[\mathbf{y}' \in S\right] + \delta. \tag{7}$$

The parameters $(\varepsilon, \delta)$ in the definition above are said to be the privacy parameters. It is important to emphasize that a differentially private mechanism $\mathcal{M}^f$ does *not*:

- Guarantee that the output $\mathcal{M}^f(\mathbf{x})$ cryptographically hides either the aggregate data $f(\mathbf{x})$ or the input record $\mathbf{x}$.
- Guarantee that the values of $\mathcal{M}^f(\mathbf{x})$ and $\mathcal{M}^f(\mathbf{x}')$ are the same when $\mathrm{dist}(\mathbf{x}, \mathbf{x}') = 1$.

A differentially private mechanism $\mathcal{M}^f$ *does* guarantee is if $\mathbf{x}, \mathbf{x}' \in X^n$ are neighboring databases (*i.e.,* $\mathrm{dist}(\mathbf{x}, \mathbf{x}') = 1$), then the distributions $\mathcal{M}^f(\mathbf{x})$ and $\mathcal{M}^f(\mathbf{x}')$ are close. Consequently, no adversary can distinguish between the cases in which it sees $\mathcal{M}^f(\mathbf{x})$ and $\mathcal{M}^f(\mathbf{x}')$, thus masking a particular user's contribution to the aggregate data. It follows that any information the adversary could possibly infer from $\mathcal{M}^f(\mathbf{x})$ can't be associated with a particular user $i \in [n]$.

All differentially private mechanisms work by introducing controlled error into the output computation; this error is generally calibrated to the sensitivity[1] of the function to be privately released. Two state-of-the-art differentially private mechanisms are the noise-based constructions of the Laplace and Gaussian mechanisms; see [15] for details on these constructions, as well as for a more complete treatment of DP.

---

[1]If $p \in \mathbb{N}$, then the $\ell_p-$sensitivity of a function $f : X^n \to Y^m$ is defined as

$$\Delta_p(f) := \max_{\substack{\mathbf{x}, \mathbf{x}' \in X^n \text{ s.t.} \\ \mathrm{dist}(\mathbf{x}, \mathbf{x}') = 1}} \left\{||f(\mathbf{x}) - f(\mathbf{x}')||_p\right\},$$

where $|| \cdot ||_p$ denotes the $\ell_p-$norm.

# 3 Client-Private Secure Aggregation

In this section, we define our model of client-private secure aggregation and describe its security model.

**Secure Aggregation.** Let $\lambda \in \mathbb{N}$ be the security parameter and $n = n(\lambda), q = q(\lambda), m = m(\lambda)$. A *secure aggregation protocol* is a secure multi-party computation (MPC) protocol executed among a set of parties $P = \{\mathcal{C}_1, \ldots, \mathcal{C}_n, \mathcal{S}\}$ consisting of $n$ clients $\mathcal{C}_1, \ldots, \mathcal{C}_n$ and a server $\mathcal{S}$. The protocol utilizes the star network graph in which each client $\mathcal{C}_i$ has an established secure communication channel with the server $\mathcal{S}$. Each client $\mathcal{C}_i$ holds a private input $\mathbf{x}_i \in \mathbb{Z}_q^m$, the server has no input, and all parties securely compute $\mathbf{z} = \sum_i \mathbf{x}_i \in \mathbb{Z}_q^m$. In each round of the protocol, every client sends a message to the server, and the server responds with a message for each client.

**Client-Private Secure Aggregation.** Here we define a novel model of secure aggregation which we call *client-private secure aggregation* (CPSA). The syntax of a CPSA protocol $\Pi$ is described in Figure 1. In CPSA, the server outputs a ciphertext $\mathbf{c}$ of the sum $\mathbf{z} = \sum_i \mathbf{x}_i$ of the clients' inputs, which only the clients can decrypt to $\mathbf{z}$. Intuitively, the security of the protocol enforces that an adversarial server learns no information about any client's input, not even the sum of their inputs. Additionally, no adversary which corrupts a subset of parties containing at least one client learns any information about any non-corrupted client's input, except for what could be inferred from the sum of the clients' inputs. We formally define the security model of CPSA in Section 3.1.

---

**Notation:** Let $\lambda \in \mathbb{N}$ be the security parameter and $n = n(\lambda), q = q(\lambda), m = m(\lambda)$, $m' = m'(\lambda) \in \mathbb{N}$. The protocol participants are $n$ clients $\mathcal{C}_1, \ldots, \mathcal{C}_n$ and a server $\mathcal{S}$. Let C be the ciphertext space of an encryption scheme.

**Input:** Each client $\mathcal{C}_i$ $(i \in [n])$ receives as input $\mathbf{x}_i \in \mathbb{Z}_q^m$; the server $\mathcal{S}$ has no input.

**Output:** The server $\mathcal{S}$ outputs a vector $\mathbf{c} \in C^{m'}$ to each client $\mathcal{C}_i$; each client $\mathcal{C}_i$ then outputs $\sum_{i=1}^{n} \mathbf{x}_i \in \mathbb{Z}_q^m$.

---

Figure 1: The syntax of a client-private secure aggregation protocol $\Pi$.

**Adding Differential Privacy.** We briefly describe a generic method to integrate differential privacy (DP) into CPSA. By adding DP to CPSA, we can enforce the additional security property that any information which an adversarial subset of parties containing at least one client can infer from the sum of the clients' inputs about *some* client's input cannot be associated with any particular client. Following the distributed DP approach outlined in [29], each client $\mathcal{C}_i$ can employ the Gaussian Mechanism $\mathcal{M}_\sigma$ $(\sigma > 0)$ to perturb their input $\mathbf{x}_i \in \mathbb{Z}_q^m$ by an independent sample $\mathbf{e}_i \leftarrow \mathsf{N}_\sigma^m$ encoded component-wise into the domain $\mathbb{Z}_q^m$ of the protocol[2], where $\mathsf{N}_\sigma$ is the Gaussian distribution with mean 0 and variance $\sigma^2$. If $\varepsilon \in (0, 1)$ and $\delta > 0$, then by Theorem 3.22 in [15], $\mathcal{M}_\sigma$ is $(\varepsilon, \delta)-$differentially privacy when $\sigma > \sqrt{\Delta_2(f)/\varepsilon} \cdot \left( \ln(25\delta/16) \right)^{1/4}$, where $\Delta_2(f)$ is the $\ell_2-$sensitivity of the sum function $f : (\mathbb{Z}_q^m)^n \to \mathbb{R}^m$ defined by $f\left((\mathbf{x}_1, \ldots, \mathbf{x}_n)\right) = \sum_i \mathbf{x}_i$. In order to obtain a reasonable bound on $\Delta_2(f)$, we can employ the norm clipping technique outlined in [30]: in PPFL, after each client $\mathcal{C}_i$ locally trains their model and obtains a vector $\mathbf{v}_i \in \mathbb{R}^m$ of model weights, they can clip $\mathbf{v}_i$ by rescaling $\mathbf{v}_i' := \mathbf{v}_i / \max\{1, \|\mathbf{v}_i\|/C\}$. It turns out that we can now bound $\Delta_2(f) \leq 2C/k$, where $C > 0$ is the clipping threshold and $k \in \mathbb{N}$ is the minimum client dataset size (Lemma 1 of [30]). Since the Gaussian distribution is additive with respect to the variance of independent samples (*i.e.*, if $e_i \leftarrow \mathsf{N}_{\sigma_i} \; \forall i \in \{1, 2\} \; (\sigma_1, \sigma_2 > 0)$, then $e_1 + e_2 \leftarrow \mathsf{N}_{\sigma'}$ where $\sigma' = \sqrt{\sigma_1^2 + \sigma_2^2}$), then each client can locally perturb its input by an independent sample

---

[2]This encoding can work, for example, by mapping each component $e_{i,j} \mapsto \lfloor (e_{i,j}/B) \cdot 2^\tau \rceil \in \mathbb{Z}_q$, where $\lfloor \cdot \rceil$ denotes the integer rounding function, $B > 0$ such that each $|e_{i,j}| \leq B$, and $\tau \in \mathbb{N}$ is a precision parameter.

$\mathbf{e}_i \leftarrow \mathsf{N}_\sigma^m$ for $\sigma > \sqrt{2C/(kn\varepsilon)} \cdot \left(\ln(25\delta/16)\right)^{1/4}$. Then, at the end of the protocol, each client outputs $\sum_i \mathbf{v}_i + \sum_i \mathbf{e}_i$, where $\sum_i \mathbf{e}_i \leftarrow \mathsf{N}_{\sigma\sqrt{n}}$ and $\sigma\sqrt{n} > \sqrt{2C/(k\varepsilon)} \cdot \left(\ln(25\delta/16)\right)^{1/4}$, as desired.

## 3.1 Security Model

We next describe the security model of CPSA. Let $\Pi$ be a CPSA protocol. We define two notions of security:

**S1:** No information about any client's input, other than the protocol output, is revealed to any other party.

**S2:** No information about any client's input, not even the protocol output, is revealed to the server.

Additionally, we are concerned with the threats:

**T1:** A subset of colluding clients attempts to learn information about another client's input.

**T2:** The server alone attempts to learn information about some client's input.

**T3:** A subset of clients collude with the server to attempt to steal information about another client's input.

Our security model of client-private secure aggregation enforces **S1** against against all threats, and **S2** against **T2**. Since by definition of client-private secure aggregation, each client computes the protocol output in the clear, then it is not possible to enforce **S2** against **T3**.

We formally prove each notion of security using the standard real/ideal world paradigm. Let $\mathcal{A}$ be a PPT adversary controlling a corrupted subset $C \subseteq P$ of parties. We define the *view* of $\mathcal{A}$ in $\Pi$ as the distribution that includes the input and random coins from each $\mathcal{P} \in C$, as well as the messages sent to each $\mathcal{P} \in C$ from the non-corrupted parties. Let Sim be a PPT simulation algorithm which simulates the view of $\mathcal{A}$ in an ideal execution of $\Pi$, without access to the non-corrupted parties' inputs. The ideal execution of $\Pi$ is defined in Figure 2. We define the following random variables:

- $\mathsf{Real}_{\Pi,P,C,\mathcal{A}}\left(1^\lambda, \{\mathbf{x}_i\}_{i \in [n]}\right)$ is the view of $\mathcal{A}$ during a real execution of $\Pi$.

- $\mathsf{Ideal}_{\Pi,P,C,\mathcal{A},\mathsf{Sim}}\left(1^\lambda, \{\mathbf{x}_i\}_{\substack{i \in [n] \text{ s.t.} \\ \mathcal{C}_i \in C}}\right)$ is the view of $\mathcal{A}$ during an ideal execution of $\Pi$.

We say that $\Pi$ is *secure against $\mathcal{A}$ controlling $C$* if there exists a PPT simulation algorithm Sim such that $\mathsf{Real}_{\Pi,P,C,\mathcal{A}}\left(1^\lambda, \{\mathbf{x}_i\}_{i \in [n]}\right) \approx_c \mathsf{Ideal}_{\Pi,P,C,\mathcal{A},\mathsf{Sim}}\left(1^\lambda, \{\mathbf{x}_i\}_{\substack{i \in [n] \text{ s.t.} \\ \mathcal{C}_i \in C}}\right)$.

There are two types of adversaries we are concerned with: semi-honest and malicious adversaries. A semi-honest adversary instructs the corrupted parties to follow the protocol honestly, but attempts to infer information about the non-corrupted parties' inputs from its view. In contrast, a malicious adversary can instruct the corrupted parties to deviate from the protocol, sending arbitrary messages or dishonestly forwarding messages to parties, to attempt to infer information about the non-corrupted parties from its view. We say that $\Pi$ is *secure in the semi-honest model* (resp., *secure in the malicious model*), if for every semi-honest (resp., malicious) adversary $\mathcal{A}$, and every subset $C \subseteq P$ of corrupted parties, $\Pi$ is secure against $\mathcal{A}$ controlling $C$.

## 4 Explicit Constructions of CPSA

In this section, we provide three explicit constructions of CPSA, and discuss their trade-offs. Actually, our first protocol construction $\Pi_0$ is just a simple modification of the two-round secure aggregation protocol of [3] described in [7]: their protocol works by the clients $\mathcal{C}_1, \ldots, \mathcal{C}_n$ computing random elements $r_1, \ldots, r_n \leftarrow \mathbb{Z}_q$, respectively, such that $\sum_i r_i = 0 \in \mathbb{Z}_q$. Each client $\mathcal{C}_i$, holding input $x_i \in \mathbb{Z}_q$, then computes a one-time pad $y_i := x_i + r_i \in \mathbb{Z}_q$ of their input and sends $y_i$ to the server $\mathcal{S}$. $\mathcal{S}$ then computes $z = \sum_i y_i = \sum_i x_i \in \mathbb{Z}_q$, and outputs $z$ to each client. The protocol $\Pi_0$ simply has

Figure 2: The ideal execution of $\Pi$.

| Criterion | $\Pi_0$ | $\Pi_1$ | $\Pi_2$ |
|---|---|---|---|
| Rounds | 2 | 3 | 3 |
| Client Computational Complexity | $\mathcal{O}(mn)$ | $\mathcal{O}(mn)$ | $\mathcal{O}(mn)$ |
| Server Computational Complexity | $\mathcal{O}(mn^2)$ | $\mathcal{O}(n(m+n))$ | $\mathcal{O}(mn^2)$ |
| Client Communication Complexity | $\mathcal{O}(mn)$ | $\mathcal{O}(m+n)$ | $\mathcal{O}(mn)$ |
| Server Communication Complexity | $\mathcal{O}(mn^2)$ | $\mathcal{O}(n(m+n))$ | $\mathcal{O}(mn^2)$ |
| Client Storage Complexity | $\mathcal{O}(m+n)$ | $\mathcal{O}(m+n)$ | $\mathcal{O}(m+n)$ |
| Server Storage Complexity | $\mathcal{O}(mn^2)$ | $\mathcal{O}(m)$ | $\mathcal{O}(m)$ |
| Reusable Decryption Key | No | No | Yes |

Table 1: A theoretical comparison of the CPSA protocols of $\Pi_0, \Pi_1$, and $\Pi_2$. The reusable decryption key criterion refers to whether the decryption key which the clients use at the end of the protocol to decrypt the encrypted sum of their inputs can be used over multiple iterations of CPSA (for PPFL). For each criterion, the protocol which is optimal with respect to this criterion is displayed in red.

each client $C_i$ use an authenticated encryption scheme to encrypt $y_i$ for every other client $C_j$ under a shared symmetric key $\mathrm{k}_{i,j}$. The server $\mathcal{S}$ simply forwards the appropriate ciphertexts to each client, who can then compute $z = \sum_i y_i = \sum_i x_i \in \mathbb{Z}_q$. We formally describe the protocol $\Pi_0$ in Appendix A.

We now construct two more CPSA protocols: $\Pi_1$ and $\Pi_2$. Table 1 provides a theoretical evaluation and comparison of the three protocols $\Pi_0, \Pi_1, \Pi_2$. $\Pi_1$ is optimal with respect to almost every criterion, except that $\Pi_0$ is two-round protocol and $\Pi_2$ supports a reusable decryption key. The latter point means that the key which the clients use at the end of the protocol to decrypt the encrypted sum of their inputs can be reused over multiple iterations of CPSA. This is important since when applied to PPFL, if the decryption key is not reusable and a client joins the collaboration in a future iteration of PPFL, in order to decrypt the encrypted initial global model from the server, a trusted setup algorithm would have to distribute the decryption key from the previous iteration of PPFL to the new client. This requires interaction between the protocol participants and the trusted setup algorithm to update the decryption key in each iteration of PPFL. On the other hand, CPSA schemes which support a reusable decryption key allow the trusted setup algorithm to simply distribute a fixed decryption key to the new client upon joining the collaboration, without the need for interaction with the protocol participants to update the decryption key after each iteration. Additionally, letting $m' \in \mathbb{N}$ be the dimension of the ciphertext vector which the server computes in the last round of CPSA, we have $m' = \mathcal{O}(mn^2)$ in $\Pi_0$, while $m' = m$ in $\Pi_1$ and $\Pi_2$. We next proceed to the explicit constructions of $\Pi_1$ and $\Pi_2$.

## 4.1 Protocol $\Pi_1$

Here, we construct a three-round CPSA protocol $\Pi_1$ with semi-honest security. At a high level, the protocol works by each client $\mathcal{C}_i$ choosing a uniformly random $s_i \leftarrow \mathbb{Z}_q$, and the clients all jointly computing $s := \sum_i s_i \leftarrow \mathbb{Z}_q$, which is kept secret from the server $\mathcal{S}$. Then, clients $\mathcal{C}_1, \ldots, \mathcal{C}_n$ compute random vectors $r_1, \ldots, r_n \leftarrow \mathbb{Z}_q$, respectively, such that $\sum_i r_i = 0 \in \mathbb{Z}_q$. Each client $\mathcal{C}_i$, holding input $x_i \in \mathbb{Z}_q$, then computes $y_i := x_i + s_i + r_i \in \mathbb{Z}_q$, and sends $y_i$ to $\mathcal{S}$. The server $\mathcal{S}$ then computes $c := \sum_i y_i = \sum_i x_i + s \in \mathbb{Z}_q$ and outputs $c$ to each client. Each client $\mathcal{C}_i$ then computes $z := c - s = \sum_i x_i \in \mathbb{Z}_q$. Note that the server computes a one-time pad $c$ of $z$ which each client can decrypt to $z$. The full protocol description is detailed in Figure 3.

---

**Setup:** All parties have access to the security parameter $\lambda \in \mathbb{N}$, a key agreement scheme $\mathsf{KA} = (\mathsf{Gen}, \mathsf{Agree})$ with key space K, a pseudorandom generator $G : \mathrm{K} \to \mathbb{Z}_q^m$, and an authenticated encryption scheme $\mathsf{AE} = (\mathsf{Gen}, \mathsf{Enc}, \mathsf{Dec})$.

**Input:** Each client $\mathcal{C}_i$ has a private input $\mathbf{x}_i \in \mathbb{Z}_q^m$; the server $\mathcal{S}$ has no input.

**Output:** The server outputs a ciphertext $\mathbf{c} \in \mathbb{Z}_q^m$ to each client $\mathcal{C}_i$ ($i \in [n]$); each client $\mathcal{C}_i$ then outputs $\sum_{i=1}^n \mathbf{x}_i \in \mathbb{Z}_q^m$.

**Round 1:**

- $\mathcal{C}_i \to \mathcal{S}$ : Generate $(\mathrm{pk}_i^{(b)}, \mathrm{sk}_i^{(b)}) \leftarrow \mathsf{KA.Gen}(1^\lambda), \forall b \in \{0, 1\}$, and output $(\mathrm{pk}_i^{(0)}, \mathrm{pk}_i^{(1)})$.

- $\mathcal{S} \to \mathcal{C}_i$ : Output $\left\{ (\mathrm{pk}_j^{(0)}, \mathrm{pk}_j^{(1)}) \right\}_{j \in [n]}$ to each client $\mathcal{C}_i$ ($i \in [n]$).

**Round 2:**

- $\mathcal{C}_i \to \mathcal{S}$ : Choose $\mathrm{s}_i \leftarrow \mathrm{K}$ and store $\mathrm{s}_i$. For all $j \in [n] \backslash \{i\}, b \in \{0, 1\}$, compute $\mathrm{k}_{i,j}^{(b)} = \mathsf{KA.Agree}(\mathrm{sk}_i^{(b)}, \mathrm{pk}_j^{(b)})$, and store $\left\{ (\mathrm{k}_{i,j}^{(0)}, \mathrm{k}_{i,j}^{(1)}) \right\}_{j \in [n] \backslash \{i\}}$. For all $j \in [n] \backslash \{i\}$, compute $c_{i,j} \leftarrow \mathsf{AE.Enc}(\mathrm{k}_{i,j}^{(0)}, \mathrm{s}_i)$. Output $\{c_{i,j}\}_{j \in [n] \backslash \{i\}}$.

- $\mathcal{S} \to \mathcal{C}_i$ : Output $\{c_{j,i}\}_{j \in [n] \backslash \{i\}}$ to each client $\mathcal{C}_i$ ($i \in [n]$).

**Round 3:**

- $\mathcal{C}_i \to \mathcal{S}$ : Receive $\{c_{j,i}\}_{j \in [n] \backslash \{i\}}$, and $\forall j \in [n] \backslash \{i\}$ compute $\mathbf{s}_j = G\left( \mathsf{AE.Dec}(\mathrm{k}_{i,j}^{(0)}, c_{j,i}) \right) \in \mathbb{Z}_q^m$. Compute $\mathbf{s}_i = G(\mathrm{s}_i)$, $\mathbf{s} = \sum_{j=1}^n \mathbf{s}_j \in \mathbb{Z}_q^m$, and store $\mathbf{s}$. For all $j \in [n] \backslash \{i\}$, compute $\mathbf{r}_{i,j} = G(\mathrm{k}_{i,j}^{(1)})$. Compute and output

$$\mathbf{y}_i = \mathbf{x}_i + \mathbf{s}_i + \sum_{j<i} \mathbf{r}_{i,j} - \sum_{j>i} \mathbf{r}_{i,j} \in \mathbb{Z}_q^m.$$

- $\mathcal{S} \to \mathcal{C}_i$ : Receive $\{\mathbf{y}_i\}_{i \in [n]}$. Compute $\mathbf{c}' = \sum_{i=1}^n \mathbf{y}_i \in \mathbb{Z}_q^m$. Output $\mathbf{c}'$ to each client $C_i$.

- $\mathcal{C}_i$ : Receive $\mathbf{c}' \in \mathbb{Z}_q^m$. Output $\mathbf{c}' - \mathbf{s} \in \mathbb{Z}_q^m$.

Figure 3: Protocol $\Pi_1$

---

**Correctness.** We now prove the correctness of $\Pi_1$, captured by Lemma 6 below.

**Lemma 6.** *After an execution of $\Pi_1$, the server computes and outputs $\mathbf{c}' = \sum_i \mathbf{x}_i + \mathbf{s} \in \mathbb{Z}_q^m$, and each client outputs $\sum_i \mathbf{x}_i \in \mathbb{Z}_q^m$.*

*Proof.* In Round 3, the server computes

$$\mathbf{c}' = \sum_i \mathbf{y}_i = \sum_i \mathbf{x}_i + \mathbf{s} + \sum_{i>j} \mathbf{r}_{i,j} - \sum_{i<j} \mathbf{r}_{i,j} \in \mathbb{Z}_q^m. \tag{8}$$

The key observation is that for each distinct pair $(i,j)$, $\mathbf{r}_{i,j} = G\Big(\mathsf{KA.Agree}(\mathrm{sk}_i^{(1)}, \mathrm{pk}_j^{(1)})\Big) = G\Big(\mathsf{KA.Agree}(\mathrm{sk}_j^{(1)}, \mathrm{pk}_i^{(1)})\Big) = \mathbf{r}_{j,i}$, hence $\sum_{i>j} \mathbf{r}_{i,j} - \sum_{i<j} \mathbf{r}_{i,j} = \mathbf{0} \in \mathbb{Z}_q^m$. We thus have that $(8) = \sum_i \mathbf{x}_i + \mathbf{s} \in \mathbb{Z}_q^m$. Each client, holding both $\mathbf{c}'$ and $\mathbf{s}$, then computes $\mathbf{c}' - \mathbf{s} = \sum_i \mathbf{x}_i \in \mathbb{Z}_q^m$. $\qquad\square$

**Security.** We now prove the security of $\mathrm{P}_1$. Specifically, we prove the security property **S1** against threats **T1** and **T3** (Lemmas 7 and 8, resp.), and security property **S2** against the threat of **T2** (Lemma 9).

**Lemma 7.** *Let $\mathcal{A}$ be a semi-honest adversary which corrupts a subset $C \subset \{\mathcal{C}_1, \ldots, \mathcal{C}_n\}$ of clients. Then, $\Pi_1$ is secure against $\mathcal{A}$ controlling $C$.*

*Proof.* Let $T = \{i \in [n] : \mathcal{C}_i \notin C\}$, and assume WLOG that $T \neq \emptyset$. Observe that in the protocol $\Pi_1$, the only messages which fall into the view of $\mathcal{A}$ that are dependent on $\{\mathbf{x}_i\}_{i \in T}$ is the computation $\mathbf{c}' = \sum_{i=1}^n \mathbf{x}_i + \mathbf{s} \in \mathbb{Z}_q^m$ sent from the server to each client in the last round of the protocol. Since each client holds $\mathbf{s} \in \mathbb{Z}_q^m$, and each corrupted client $\mathcal{C}_i \in C$ holds their input $\mathbf{x}_i \in \mathbb{Z}_q^m$, then nothing more than $\mathbf{z} := \sum_{i \in T} \mathbf{x}_i = \mathbf{c}' - \sum_{i \notin T} \mathbf{x}_i - \mathbf{s} \in \mathbb{Z}_q^m$ is revealed to $\mathcal{A}$. So, the simulation algorithm $\mathsf{Sim}$ which is the same as $\Pi_1$, except that each non-corrupted client $\mathcal{C}_j$ is given input $\mathbf{x}_j' \leftarrow \mathbb{Z}_q^m$ such that $\sum_{j \in T} \mathbf{x}_j = \mathbf{z} \in \mathbb{Z}_q^m$, is such that the view of $\mathcal{A}$ in this simulation is distributed identically to $\mathsf{Real}_{\Pi_1, P, C, \mathcal{A}}\Big(1^\lambda, \{\mathbf{x}_i\}_{i \in [n]}\Big)$. Hence we have $\mathsf{Real}_{\Pi_1, P, C, \mathcal{A}}\Big(1^\lambda, \{\mathbf{x}_i\}_{i \in [n]}\Big) \equiv \mathsf{Ideal}_{\Pi_1, P, C, \mathcal{A}, \mathsf{Sim}}\Big(1^\lambda, \{\mathbf{x}_i\}_{i \notin T}, \mathbf{z}\Big)$, which completes the proof. $\qquad\square$

**Lemma 8.** *Let $\mathcal{A}$ be a semi-honest adversary which corrupts a subset $C \subset P$ of parties containing the server $\mathcal{S}$ and at least one client. Then, $\Pi_1$ is secure against $\mathcal{A}$ controlling $C$.*

*Proof.* Let $\mathsf{Real}_{\Pi_1, P, C, \mathcal{A}}\Big(1^\lambda, \{\mathbf{x}_i\}_{i \in [n]}\Big)$ denote the distribution of the view of $\mathcal{A}$ in a real execution of $\Pi_1$ in which $\mathcal{A}$ corrupts $C$. We'll construct a PPT simulation algorithm $\mathsf{Sim}$ which simulates the view of $\mathcal{A}$ without access to the non-corrupted clients' inputs. By definition of the ideal execution of $\Pi_1$ (Figure 2), this completes the proof. Let $T = \{i \in [n] : C_i \notin C\}$. We may assume WLOG that $T \neq \emptyset$.

We'll actually make one small modification to the ideal execution of $\Pi_1$ in this case. Although the simulation algorithm $\mathsf{Sim}$ is not given access to the non-corrupted parties' inputs, we will endow $\mathsf{Sim}$ with the sum $\mathbf{z} := \sum_{i \in T} \mathbf{x}_i + \sum_{i \in T} \mathbf{s}_i \in \mathbb{Z}_q^m$ of the non-corrupted parties' inputs. Note that this is without loss of generality since any adversary, given the server's output $\sum_{i \in [n]} \mathbf{x}_i + \mathbf{s} \in \mathbb{Z}_q^m$, and the values $\big\{(\mathbf{x}_i, \mathbf{s}_i)\big\}_{i \notin T} \subseteq \mathbb{Z}_q^m \times \mathbb{Z}_q^m$ held by the corrupted parties can efficiently compute $\mathbf{z}$. We thus replace $\mathsf{Ideal}_{\Pi_1, P, C, \mathcal{A}, \mathsf{Sim}}\Big(1^\lambda, \{\mathbf{x}_i\}_{i \notin T}\Big)$ with $\mathsf{Ideal}_{\Pi_1, P, C, \mathcal{A}, \mathsf{Sim}}\Big(1^\lambda, \{\mathbf{x}_i\}_{i \notin T}, \mathbf{z}\Big)$.

We now proceed by a standard hybrid argument:

- $\mathcal{H}_0$ : This hybrid is simply a real execution of $\Pi_1$.

- $\mathcal{H}_1$ : This hybrid is the same as $\mathcal{H}_0$ except that in Round 3, for each $\mathcal{C}_i \in \{\mathcal{C}_1, \ldots, \mathcal{C}_n\} \backslash C$, we choose $\mathbf{r}_i \leftarrow \mathbb{Z}_q^m$ such that $\sum_{i \in T} \mathbf{r}_i = \mathbf{z} \in \mathbb{Z}_q^m$, and $C_i$ instead lets $\mathbf{y}_i := \mathbf{r}_i \in \mathbb{Z}_q^m$. By Lemma 6.1 in [3], we have that $\mathcal{H}_0 \equiv \mathcal{H}_1$.

We define $\mathsf{Sim}$ by $\mathcal{H}_1$, and it follows that $\mathsf{Real}_{\Pi_1, P, C, \mathcal{A}}\Big(1^\lambda, \{\mathbf{x}_i\}_{i \in [n]}\Big) \equiv \mathcal{H}_0 \equiv \mathcal{H}_1 \equiv \mathsf{Ideal}_{\Pi_1, P, C, \mathcal{A}, \mathsf{Sim}}\Big(1^\lambda, \{\mathbf{x}_i\}_{i \notin T}, \mathbf{z}\Big)$. $\qquad\square$

**Lemma 9.** *Let $\mathcal{A}$ be a semi-honest adversary which corrupts the server $\mathcal{S}$. Then, $\Pi_1$ is secure against $\mathcal{A}$ controlling $C = \{\mathcal{S}\}$.*

*Proof.* Let Sim be the simulation algorithm which is the same as $\Pi_1$, except that in Round 3, each client $\mathcal{C}_i$ chooses $\mathbf{u}_i \leftarrow \mathbb{Z}_q^m$ and instead sends $\mathbf{y}_i := \mathbf{u}_i + \mathbf{s}_i \in \mathbb{Z}_q^m$ to the server. Since the distribution of each $\mathbf{s}_i$ is computationally indistinguishable from the uniform distribution on $\mathbb{Z}_q^m$, it follows that

$$\mathsf{Real}_{\Pi_1, P, C, \mathcal{A}}\left(1^\lambda, \{\mathbf{x}_i\}_{i \in [n]}\right) \approx_c \mathsf{Ideal}_{\Pi_1, P, C, \mathcal{A}, \mathsf{Sim}}\left(1^\lambda, \{\mathbf{x}_i\}_{i \notin T}, \mathbf{z}\right).$$
□

## 4.2 Protocol $\Pi_2$

We now construct a second novel CPSA protocol $\Pi_2$ which achieves semi-honest security. While the computational and communication complexity of $\Pi_2$ is greater than $\Pi_1$, in $\Pi_2$ the server computes a ciphertext of the sum of the clients' inputs under an asymmetric key pair shared by the clients, which can be reused over multiple iterations of CPSA for PPFL. At a high level, the protocol works by each client $\mathcal{C}_i$ first receiving a public/secret key pair $(\mathrm{pk}, \mathrm{sk})$ for an additive homomorphic encryption (AHE) scheme from a trusted setup algorithm. Next, each client $\mathcal{C}_i$ splits their input $x_i \in \mathbb{Z}_q$ into $n$ additive secret shares $\{s_{i,j}\}_{j \in [n]} \subseteq \mathbb{Z}_q$, one for every client, and distributes each share $\mathbf{s}_{i,j}$ to Client $\mathcal{C}_j$ by way of the server. Now, each client $\mathcal{C}_i$ holds shares $\{s_{j,i}\}_{j \in [n]} \subseteq \mathbb{Z}_q^m$, and sums over the shares to compute $t_i = \sum_{j \in [n]} s_{j,i} \in \mathbb{Z}_q$, which is a share of the sum $z = \sum_{r \in [n]} x_r \in \mathbb{Z}_q$. Each client $\mathcal{C}_i$ then uses the AHE scheme to encrypt their share $t_i$ under $\mathrm{pk}$ to obtain a ciphertext $c_i$ which they send to the server. The server homomorphically reconstructs $z$ by homomorphically adding $\{c_i\}_{i \in [n]}$, obtaining a ciphertext $c'$ of $z$. The server then outputs $c'$ to each client, which uses $\mathrm{sk}$ to decrypt $c'$ to $z$. The full protocol description is detailed in Figure 4.

**Remark.** Note that in our protocol $\Pi_2$, each client $\mathcal{C}_i$ holds the same AHE secret key $\mathrm{sk}$. Although this is not typical in homomorphic encryption solutions to MPC when the number of parties is greater than 2, in this application the AHE scheme is used to hide the protocol output, which can be viewed as a secret shared by all clients, from the server. If an adversary controlling the server corrupts a client, then the AHE secret key $\mathrm{sk}$ falls into the view of the adversary. Thus the adversary learns all of the clients' shares of the sum $\mathbf{z} \in \mathbb{Z}_q^m$, and hence the plaintext sum $\mathbf{z}$ falls into the adversary's view. But, since the adversary has corrupted the client, then $\mathbf{z}$ already falls into its view, and so no further information about any non-corrupted client's input is revealed to the adversary.

**Correctness.** We now prove the correctness of $\Pi_2$, captured by Lemma 10 below.

**Lemma 10.** *After an execution of $\Pi_2$, the server outputs a vector of ciphertexts $\mathbf{c}'' \in \mathrm{C}^m$ to each client, and each client outputs $\sum_i \mathbf{x}_i \in \mathbb{Z}_q^m$.*

*Proof.* After the end of Round 1, each client $\mathcal{C}_i$ holds their secret key $\mathrm{sk}_i$ and a public key $\mathrm{pk}_j$ from every other client $\mathcal{C}_j$ ($j \neq i$) for the key agreement scheme KA. So, in Round 2, each client $\mathcal{C}_i$ computes a shared symmetric key $\mathrm{k}_{i,j}$ with every other client $\mathcal{C}_j$. $\mathcal{C}_i$ then splits its private input into additive secret shares $\{\mathbf{s}_{i,j}\}_{j \in [n]} \subseteq \mathbb{Z}_q^m$ for every client, encrypts each $\mathcal{C}_j$'s share $\mathbf{s}_{i,j}$ ($j \neq i$) with the authenticated encryption scheme AE under the shared symmetric key $\mathrm{k}_{i,j}$, and sends the resulting ciphertexts to the server. The server then forwards to each client $\mathcal{C}_i$ ciphertexts of its shares $\mathbf{s}_{j,i}$ from every other client $\mathcal{C}_j$, which it decrypts in Round 3 to obtain $\{\mathbf{s}_{j,i}\}_{j \in [n]}$. $\mathcal{C}_i$ then computes $\mathbf{t}_i = \sum_{j \in [n]} \mathbf{s}_{j,i} \in \mathbb{Z}_q^m$, which it follows is a share of $\mathbf{z} := \sum_{j \in [n]} \mathbf{x}_i \in \mathbb{Z}_q^m$. Finally, $\mathcal{C}_i$ uses the additive homomorphic encryption scheme AHE to encrypt $\mathbf{t}_i$ under $\mathrm{pk}$, obtaining a ciphertext $\mathbf{c}'_i$, which it sends to the server. By definition of additive secret sharing, it follows that $\sum_{i \in [n]} \mathbf{t}_i = \mathbf{z} \in \mathbb{Z}_q^m$. So, the server, each holding AHE ciphertext vectors $\mathbf{c}'_1, \ldots, \mathbf{c}'_n$ of $\mathbf{t}_1, \ldots, \mathbf{t}_n$, respectively, component-wise homomorphically adds $\{\mathbf{c}'_1, \ldots, \mathbf{c}'_n\}$ to obtain a ciphertext $\mathbf{c}''$ of $\mathbf{z}$, which it outputs to each client. Each client then computes $\mathbf{z} = \mathsf{AHE.Dec}(\mathrm{sk}, \mathbf{c}'')$. □

**Security.** We now prove that $\Pi_2$ is secure with respect to **S1** and **S2** in the semi-honest model. Let $\mathcal{A}$ be a semi-honest adversary controlling a subset $C \subseteq P$ of corrupted parties. First, we note

**Setup:** All parties have access to the security parameter $\lambda \in \mathbb{N}$, a key agreement scheme $\mathsf{KA} = (\mathsf{Gen}, \mathsf{Agree})$ with key space $\mathrm{K}$, an authenticated encryption scheme $\mathsf{AE} = (\mathsf{Gen}, \mathsf{Enc}, \mathsf{Dec})$, and an additive homomorphic encryption scheme $\mathsf{AHE} = (\mathsf{Gen}, \mathsf{Enc}, \mathsf{Dec}, \mathsf{Add})$ with plaintext space $\mathbb{Z}_q$ and ciphertext space $\mathrm{C}$. As part of a one-time trusted setup phase, a trusted setup algorithm generates $(\mathrm{pk}, \mathrm{sk}) \leftarrow \mathsf{AHE.Gen}(1^\lambda)$ and sends $(\mathrm{pk}, \mathrm{sk})$ to each client $C_i$ ($i \in [n]$).

**Input:** Each client $C_i$ has a private input $\mathbf{x}_i \in \mathbb{Z}_q^m$; the server $S$ has no input.

**Output:** The server outputs a ciphertext $\mathbf{c}'' \in \mathrm{C}^m$ to each client $C_i$ ($i \in [n]$); each client then outputs $\sum_{i=1}^n \mathbf{x}_i \in \mathbb{Z}_q^m$.

**Round 1:**

- $C_i \to S$ : Generate $(\mathrm{pk}_i, \mathrm{sk}_i) \leftarrow \mathsf{KA.Gen}(1^\lambda)$, and output $\mathrm{pk}_i$.

- $S \to C_i$ : Output $\{\mathrm{pk}_j\}_{j=1}^n$.

**Round 2:**

- $C_i \to S$ : For each $j \in [n]\backslash\{i\}$, compute $\mathrm{k}_{i,j} = \mathsf{KA.Agree}(\mathrm{sk}_i, \mathrm{pk}_j)$. For all $j \in [n-1]$, choose $\mathbf{s}_{i,j} \leftarrow \mathbb{Z}_q^m$, and let $\mathbf{s}_{i,n} = \mathbf{x}_i - \sum_{j \in [n-1]} \mathbf{s}_{i,j} \in \mathbb{Z}_q^m$.

For each $j \in [n]\backslash\{i\}$, perform the following:
For all $k \in [m]$, compute $c_{i,j,k} \leftarrow \mathsf{AE.Enc}(\mathrm{k}_{i,j}, s_{i,j,k})$, and let $\mathbf{c}_{i,j} = (c_{i,j,k})_{k \in [m]}$.
Output $\{\mathbf{c}_{i,j}\}_{j \in [n]\backslash\{i\}}$.

- $S \to C_i$ : Receive $\{\mathbf{c}_{i,j}\}_{j \in [n]\backslash\{i\}}$ from each client $C_i$ ($i \in [n]$). Output $\{\mathbf{c}_{j,i}\}_{j \in [n]\backslash\{i\}}$ to each client $C_i$.

**Round 3:**

- $C_i \to S$ : For each $j \in [n]\backslash\{i\}$, perform the following:
For all $k \in [m]$, compute $s_{j,i,k} = \mathsf{AE.Dec}(\mathrm{k}_{i,j}, c_{j,i,k})$, and let $\mathbf{s}_{j,i} = (s_{j,i,k})_{k \in [m]}$.
Compute $\mathbf{t}_i = \sum_{j \in [n]} \mathbf{s}_{j,i} \in \mathbb{Z}_q^m$. For each $k \in [m]$, compute $c'_{i,k} \leftarrow \mathsf{AHE.Enc}(\mathrm{pk}, t_{i,k})$.
Output $\mathbf{c}'_i := (c'_{i,k})_{k \in [m]} \in \mathrm{C}^m$.

- $S \to C_i$ : Initialize $\mathbf{c}'' := \mathbf{c}'_1$. For all $i \in \{2, \ldots, n\}$, perform the following:
For all $k \in [m]$, update $c''_k \leftarrow \mathsf{AHE.Add}(c''_k, c'_{i,k})$.
Output $\mathbf{c}''$ to each client $C_i$.

- $C_i$ : Receive $\mathbf{c}''$. For all $k \in [m]$, compute $z_k = \mathsf{AHE.Dec}(\mathrm{sk}, c''_k)$.
Output $\mathbf{z} := (z_k)_{k \in [m]} \in \mathbb{Z}_q^m$.

Figure 4: Protocol $\Pi_2$

that the security **S2** of $\Pi_2$ when $C = \{\mathcal{S}\}$ follows immediately from the semantic security of the authenticated encryption and additive homomorphic encryption schemes. It thus suffices to prove the security **S1** of $\Pi_2$ when there exists some client $\mathcal{C}_i \in C$. Lemma 11 below completes the proof of security.

**Lemma 11** (Security). *Let $\mathcal{A}$ be a semi-honest adversary which corrupts a subset $C \subset P$ of parties containing at least one client. Then, $\Pi_2$ is secure against $\mathcal{A}$ controlling $C$.*

*Proof.* We may assume without loss of generality that $T := \{i \in [n] : \mathcal{C}_i \notin C\} \neq \emptyset$. Let $\mathbf{z} = \sum_{i \in T} \mathbf{x}_i \in \mathbb{Z}_q^m$. We'll actually make one small modification to the ideal execution of $\Pi_2$ in this case. Although the simulation algorithm $\mathsf{Sim}$ is not given access to the non-corrupted parties' inputs, we will endow $\mathsf{Sim}$ with the sum $\mathbf{z} := \sum_{i \in T} \mathbf{x}_i \in \mathbb{Z}_q^m$ of the non-corrupted parties' inputs. Note that this is without loss of generality since any adversary, given the protocol output $\sum_{i \in [n]} \mathbf{x}_i \in \mathbb{Z}_q^m$ and the corrupted parties inputs $\{\mathbf{x}_i\}_{i \notin T} \subseteq \mathbb{Z}_q^m$ can efficiently compute $\mathbf{z}$. We thus replace $\mathsf{Ideal}_{\Pi_2, P, C, \mathcal{A}, \mathsf{Sim}}\left(1^\lambda, \{\mathbf{x}_i\}_{i \notin T}\right)$ with $\mathsf{Ideal}_{\Pi_2, P, C, \mathcal{A}, \mathsf{Sim}}\left(1^\lambda, \{\mathbf{x}_i\}_{i \notin T}, \mathbf{z}\right)$. We now proceed by a standard hybrid argument.

|  | **Client** |  |  | **Server** |  |  |
|---|---|---|---|---|---|---|
| $n$ | $\Pi_0$ | $\Pi_1$ | $\Pi_2$ | $\Pi_0$ | $\Pi_1$ | $\Pi_2$ |
| 50 | 0.458 | 0.243 | 16.763 | 0.000690 | 0.00191 | 0.195 |
| 100 | 0.934 | 0.482 | 16.990 | 0.00659 | 0.00497 | 0.415 |
| 250 | 2.334 | 1.234 | 18.0570 | 0.0342 | 0.0198 | 1.0150 |

Table 2: Client and server running times for $n \in \{50, 100, 250\}$ ($m = 100$).

| **Client** |  |  |
|---|---|---|
|  | **Comm Overhead (KB)** | **Space Overhead (KB)** |
| $\Pi_0$ | 868.029 | 21.303 |
| $\Pi_1$ | 14.099 | 29.411 |
| $\Pi_2$ | 925.239 | 21.391 |
| **Server** |  |  |
|  | **Comm Overhead (MB)** | **Space Overhead (MB)** |
| $\Pi_0$ | 87.751 | 86.791 |
| $\Pi_1$ | 2.3603 | 0.0044 |
| $\Pi_2$ | 92.996 | 5.720 |

Table 3: Client and server communication and space overhead ($n = 100, m = 100$).

- $\mathcal{H}_0$ : This hybrid is simply a real execution of $\Pi_2$.

- $\mathcal{H}_1$ : This hybrid is the same as $\mathcal{H}_0$, except that for each non-corrupted client $\mathcal{C}_i \in \{\mathcal{C}_1, \ldots, \mathcal{C}_n\} \backslash C$, in Round 2, for each $\mathcal{C}_j \in C$, we set $\mathbf{s}_{i,j} \leftarrow \mathbb{Z}_q^m$. Since the view of the adversary after Round 2 contains $\{\mathbf{s}_{i,j}\}_{\substack{i \in T \\ j \notin T}}$, then by the security of the additive secret sharing scheme we have that $\mathcal{H}_0 \equiv \mathcal{H}_1$.

- $\mathcal{H}_2$ : In this hybrid, it will be more convenient to index the non-corrupted clients by $\mathcal{C}_{i_1}, \ldots, \mathcal{C}_{i_r}$. For each $j \in [r-1]$, in Round 2, $\mathcal{C}_{i_j}$ generates shares $\{\mathbf{s}_{i_j,t}\}_{t \in [n]}$ of 0, while $\mathcal{C}_{i_r}$ generates shares $\{\mathbf{s}_{i_r,t}\}_{t \in [n]}$ of $\mathbf{z}'$. Similarly, by the security of the additive secret sharing scheme we have that $\mathcal{H}_1 \equiv \mathcal{H}_2$.

We define Sim by $\mathcal{H}_2$, and it follows that $\mathsf{Real}_{\Pi_2,P,C,\mathcal{A}}(1^\lambda, \{\mathbf{x}_i\}_{i \in [n]}) \equiv \mathcal{H}_0 \equiv \mathcal{H}_1 \equiv \mathcal{H}_2 \equiv \mathsf{Ideal}_{\Pi_2,P,C,\mathcal{A},\mathsf{Sim}}\left(1^\lambda, \{\mathbf{x}_i\}_{i \notin T}, \mathbf{z}\right).$ $\qquad\square$

## 5 Experimental Results

In this section, we empirically evaluate our CPSA protocols $\Pi_0, \Pi_1, \Pi_2$ with respect to their running time, communication overhead, and space overhead. We implemented both protocols in Python, using Elliptic Curve Diffie-Hellman for a key agreement scheme, AES-GCM for an authenticated encryption scheme, AES-CTR for a pseudorandom generator, and Paillier Encryption [26] for an additive homomorphic encryption scheme. For each protocol construction, we conducted the following experiments:

- Measure running time for client and server for number $n$ of clients, dimension $m$ of clients' input vector, and modulus $q$, when $n \in \{50, 100, 250\}$, $m = 100$, $q = 2^{128}$.

- Measure communication and space overhead for client and server when $n = 100, m = 100, q = 2^{128}$.

All experiments were run on a MacBook Pro with Intel Core i7 6-core 2.6 GHz CPU, and each party was simulated as a sub-process. Our experiments only measure the local performance of the protocol, and in particular ignore network latency. Table 2 compares the running times vs. number $n \in \{50, 100, 250\}$ of clients between the three protocols for the client and server, respectively. Our experiments indicate that the running times of $\Pi_1$ are most performant. We can see that while our theoretical analysis of the computational complexity of protocols $\Pi_2$ and $\Pi_0$ indicates they are

identical, in reality the running time of $\Pi_2$ for both the client and server is noticeably higher than that of $\Pi_0$. This is due to the cost of the homomorphic operations of Paillier Encryption. However, note that the running times for $\Pi_2$ are still practical, with the client and server obtaining running times less than 19 s and 2 s, respectively, even when scaled to 250 clients.

Table 3 displays the client and server communication and space overhead for each protocol construction when $n = 100$ and $m = 100$. Again, our experiments indicate that $\Pi_1$ is generally most performant with respect to both communication and space overhead; this confirms our theoretical analysis (Table 1) which shows that $\Pi_1$ achieves linear communication complexity. Between $\Pi_0$ and $\Pi_2$, we remark that for both the client and server, the communication and space overhead are quite comparable, except that the server's space overhead of $5.72$ MB in $\Pi_2$ is significantly lower than for $\Pi_0$ ($86.791$ MB). This is because in $\Pi_2$, the server outputs to each client a single $m-$dimensional vector of ciphertexts, while in $\Pi_0$, the server outputs to each client an $(n-1)-$sized set of $m-$dimensional ciphertext vectors.

## 6   Conclusions and Future Work

In this work, we propose a novel model of secure aggregation, called client-private secure aggregation (CPSA), in which the server computes an encrypted global model that can only be decrypted by the clients. When composed with differential privacy, the security property is that no party learns any information beyond the global model about any client's dataset, and any possibly inferred information from the global model cannot be associated with any particular client. In particular, the server learns no information about any client's dataset, not even the global model. We provide three explicit constructions of CPSA, each with varying trade-offs, and prove correctness and security in the semi-honest model for each construction. Finally, we empirically evaluate our three constructions to demonstrate their practicality and illustrate the trade-offs offered by each protocol.

There are several elements of future work which we seek to incorporate into the full version of this work. First, we believe it's possible to prove that (a simple modification to) each of these protocols is secure against a malicious adversary. Also, we believe we can employ techniques to mitigate the client and server running time of our implementation of $\Pi_2$. For example, it may be possible to use the additive homomorphic version of ElGamal Encryption [17], which works over small input domains, and is more computationally efficient than Paillier Encryption. Alternatively, we wish to investigate packing Paillier ciphertexts, following [25], to improve the client and server running times.

**Acknowledgements.**   We thank Xianrui Meng for fruitful conversations and feedback on this work. Additionally, we thank the anonymous reviewers of FL-NeurIPS'22 for their helpful suggestions and feedback.

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

# A   Protocol $\Pi_0$

The protocol $\Pi_0$, described below, achieves a client-private secure aggregation protocol with semi-honest security through a simple modification to the two-round semi-honest secure variant of [3] (the two-round variant is described in [7]). At a high level, the protocol works by each client $\mathcal{C}_i$ computing a quasi-one-time pad of its private input $\mathbf{x}_i \in \mathbb{Z}_q^m$ as $\mathbf{y}_i := \mathbf{x}_i + \mathbf{r}_i \in \mathbb{Z}_q^m$, where the clients' random masks $\mathbf{r}_1, \ldots, \mathbf{r}_n \leftarrow \mathbb{Z}_q^m$ are chosen such that $\sum_i \mathbf{r}_i = \mathbf{0} \in \mathbb{Z}_q^m$. Each client $C_i$ then simply encrypts $\mathbf{y}_i$ for every other client $\mathcal{C}_j$ under a shared symmetric key $\mathrm{k}_{i,j}$, and sends the ciphertexts to the server, which routes them to the appropriate clients. Each client $\mathcal{C}_i$ then has a set of one-time pads $\{\mathbf{y}_j\}_{j \in [n]}$ whose random masks sum to zero, hence the client computes $\mathbf{z} := \sum_{j \in [n]} \mathbf{y}_j = \sum_{j \in [n]} \mathbf{x}_j$, as desired.

Figure 4 contains the full protocol description of $\Pi_0$.

---

**Setup:** All parties have access to the security parameter $\lambda \in \mathbb{N}$, a key agreement scheme $\mathsf{KA} = (\mathsf{Gen}, \mathsf{Agree})$ with key space $\mathrm{K}$, a pseudorandom generator $G : \mathrm{K} \to \mathbb{Z}_q^m$, and an authenticated encryption scheme $\mathsf{AE} = (\mathsf{Gen}, \mathsf{Enc}, \mathsf{Dec})$ with ciphertext space $\mathrm{C}$.

**Input:** Each client $\mathcal{C}_i$ has a private input $\mathbf{x}_i \in \mathbb{Z}_q^m$; the server $\mathcal{S}$ has no input.

**Output:** The server outputs ciphertexts $(\mathbf{c}_{j,i})_{j \in [n] \setminus \{i\}} \in \mathrm{C}^{m(n-1)}$ to each client $\mathcal{C}_i$ $(i \in [n])$; each client then outputs $\sum_{i=1}^{n} \mathbf{x}_i \in \mathbb{Z}_q^m$.

**Round 1:**

• $\mathcal{C}_i \to \mathcal{S}$ : Generate $(\mathrm{pk}_i^{(b)}, \mathrm{sk}_i^{(b)}) \leftarrow \mathsf{KA.Gen}(1^\lambda), \forall b \in \{0,1\}$, and output $(\mathrm{pk}_i^{(0)}, \mathrm{pk}_i^{(1)})$.

• $\mathcal{S} \to \mathcal{C}_i$ : Output $\left\{ (\mathrm{pk}_j^{(0)}, \mathrm{pk}_j^{(1)}) \right\}_{j=1}^{n}$.

**Round 2:**

• $\mathcal{C}_i \to \mathcal{S}$ : For all $j \in [n] \setminus \{i\}, b \in \{0,1\}$, compute $\mathrm{k}_{i,j}^{(b)} = \mathsf{KA.Agree}(\mathrm{sk}_i^{(b)}, \mathrm{pk}_j^{(b)})$. For all $j \in [n] \setminus \{i\}$, compute $\mathbf{r}_{i,j} = G(\mathrm{k}_{i,j}^{(1)})$. Let $\mathbf{y}_i = \mathbf{x}_i + \sum_{j<i} \mathbf{r}_{i,j} - \sum_{j>i} \mathbf{r}_{i,j} \in \mathbb{Z}_q^m$.

For all $j \in [n] \setminus \{i\}, k \in [m]$, compute $c_{i,j,k} \leftarrow \mathsf{AE.Enc}(\mathrm{k}_{i,j}^{(0)}, y_{i,k})$. For all $j \in [n] \setminus \{i\}$, let $\mathbf{c}_{i,j} = (c_{i,j,k})_{k \in [m]} \in \mathrm{C}^m$. Output $\{\mathbf{c}_{i,j}\}_{j \in [n] \setminus \{i\}}$.

• $\mathcal{S} \to \mathcal{C}_i$ : Store $\left\{ \{\mathbf{c}_{j,i}\}_{j \in [n] \setminus \{i\}} \right\}_{i \in [n]}$. Output $\{\mathbf{c}_{j,i}\}_{j \in [n] \setminus \{i\}}$ to each client $\mathcal{C}_i$.

• $\mathcal{C}_i$ : For all $j \in [n] \setminus \{i\}, k \in [m]$, compute $w_{j,k} = \mathsf{AE.Dec}(\mathrm{k}_{i,j}^{(0)}, c_{j,i,k})$. For all $j \in [n]$, let $\mathbf{w}_j = (w_{j,k})_{k \in [m]} \in \mathbb{Z}_q^m$ if $j \neq i$, or $\mathbf{w}_j = \mathbf{y}_i$ otherwise. Output $\mathbf{z} = \sum_{j \in [n]} \mathbf{w}_j \in \mathbb{Z}_q^m$.

---

Figure 5:  Protocol $\Pi_0$

**Correctness.**   We prove the correctness of $\Pi_0$ in Lemma 12 below.

**Lemma 12** (Correctness)**.** *After an execution of $\Pi_0$, the server outputs to each client $\mathcal{C}_i$ $(i \in [n])$ ciphertexts $\{\mathbf{c}_{j,i}\}_{j \in [n] \setminus \{i\}} \in \mathrm{C}^{m(n-1)}$, and each client outputs $\sum_i \mathbf{x}_i \in \mathbb{Z}_q^m$.*

*Proof.* In Round 1, each client $\mathcal{C}_i$ uses the key agreement scheme to generate two sets of public/secret key pairs $\left( (\mathrm{pk}_i^{(b)}, \mathrm{sk}_i^{(b)}) \right)_{b \in \{0,1\}}$, and sends $(\mathrm{pk}_i^{(0)}, \mathrm{pk}_i^{(1)})$ to the server. The server then forwards $(\mathrm{pk}_i^{(0)}, \mathrm{pk}_i^{(1)})_{i \in [n]}$ to each client. For each pair of clients $(\mathcal{C}_i, \mathcal{C}_j)$ $(i \neq j)$, and for each $b \in \{0,1\}$, $\mathcal{C}_i$ (resp., $\mathcal{C}_j$) uses their secret key $\mathrm{sk}_i^{(b)}$ (resp., $\mathrm{sk}_j^{(b)}$) and the public key $\mathrm{pk}_j^{(b)}$ (resp., $\mathrm{pk}_i^{(b)}$) of client $\mathcal{C}_j$ (resp., $\mathcal{C}_i$) to compute a shared random key $\mathrm{k}_{i,j}^{(b)} = \mathrm{k}_{j,i}^{(b)}$.

Now, each client $\mathcal{C}_i$ computes $\mathbf{r}_{i,j} = G(\mathrm{k}_{i,j}^{(1)})$, $\forall j \in [n]\backslash\{i\}$, $\mathbf{y}_i = \mathbf{x}_i + \sum_{j<i} \mathbf{r}_{i,j} - \sum_{j>i} \mathbf{r}_{i,j} \in \mathbb{Z}_q^m$, uses the authenticated encryption scheme to encrypt each component of $\mathbf{y}_i$ under $\mathrm{k}_{i,j}^{(0)}$ to obtain a vector of ciphertexts $\mathbf{c}_{i,j}$, $\forall j \in [n]\backslash\{i\}$, and outputs $\{\mathbf{c}_{i,j}\}_{j\in[n]\backslash\{i\}}$ to the server. The server forwards $\{\mathbf{c}_{j,i}\}_{j\in[n]\backslash\{i\}}$ to each client $\mathcal{C}_i$.

Now, each client $\mathcal{C}_i$ uses the authenticated encryption scheme to decrypt the components of each $\mathbf{c}_{j,i}$, using $\mathrm{k}_{i,j}^{(0)}$, to recover $\mathbf{y}_j \in \mathbb{Z}_q^m$, $\forall j \in [n]\backslash\{i\}$. $\mathcal{C}_i$ then computes $\mathbf{z} = \sum_{j\in[n]} \mathbf{y}_j = \sum_{j\in[n]} \mathbf{x}_j + \sum_{j<k} \mathbf{r}_{j,k} + \sum_{j>k} \mathbf{r}_{j,k} = \sum_{j\in[n]} \mathbf{x}_j + \sum_{j<k} \mathbf{r}_{j,k} - \sum_{j>k} \mathbf{r}_{k,j} = \sum_{j\in[n]} \mathbf{x}_j$, since each $\mathbf{r}_{j,k} = \mathbf{r}_{k,j}$. $\qquad\square$

**Security.** We now prove that $\Pi_0$ is secure with respect to **S1** and **S2** in the semi-honest model. Let $\mathcal{A}$ be a semi-honest adversary controlling a subset $C \subseteq P$ of corrupted parties. First, we note that the security **S2** of $\Pi_0$ when $C = \{\mathcal{S}\}$ follows immediately from the semantic security of the authenticated encryption scheme. So, it suffices to prove the security **S1** of $\Pi_0$ when there exists some client $\mathcal{C}_i \in C$. Lemma 13 below completes the security proof.

**Lemma 13** (Security). *Let $\mathcal{A}$ be a semi-honest adversary which corrupts a subset $C \subseteq P$ of parties such that some client $\mathcal{C}_i \in C$. Then, $\Pi_0$ is secure against $\mathcal{A}$ controlling $C$.*

*Proof.* Let $C \subseteq P$, and $\mathsf{Real}_{\Pi_0,P,C,\mathcal{A}}(1^\lambda, \{\mathbf{x}_i\}_{i\in[n]})$ denote the distribution of the view of $\mathcal{A}$ in a real execution of $\Pi_0$ in which $\mathcal{A}$ corrupts $C$. We'll construct a PPT simulation algorithm Sim which simulates the view of $\mathcal{A}$ without access to the non-corrupted clients' inputs. By definition of the ideal execution of $\Pi_0$ (Figure 2), this completes the proof. Let $T = \{i \in [n] : \mathcal{C}_i \notin C\}$. We may assume WLOG that $T \neq \emptyset$.

Just as in the proof of Lemma 11, we may endow Sim with the sum $\mathbf{z} := \sum_{i\in T} \mathbf{x}_i \in \mathbb{Z}_q^m$ of the non-corrupted parties' inputs. We thus replace $\mathsf{Ideal}_{\Pi_0,P,C,\mathcal{A},\mathsf{Sim}}\left(1^\lambda, \{\mathbf{x}_i\}_{i\notin T}\right)$ with $\mathsf{Ideal}_{\Pi_0,P,C,\mathcal{A},\mathsf{Sim}}\left(1^\lambda, \{\mathbf{x}_i\}_{i\notin T}, \mathbf{z}\right)$.

We now proceed by a standard hybrid argument.

- $\mathcal{H}_0$ : This hybrid is simply a real execution of $\Pi_0$.

- $\mathcal{H}_1$ : For each $\mathcal{C}_i \in \{\mathcal{C}_1, \ldots, \mathcal{C}_n\}\backslash C$, we choose $\mathbf{r}_i \leftarrow \mathbb{Z}_q^m$ such that $\sum_{i\in T} \mathbf{r}_i = \mathbf{z} \in \mathbb{Z}_q^m$, and $\mathcal{C}_i$ instead lets $\mathbf{y}_i := \mathbf{r}_i \in \mathbb{Z}_q^m$. Note that since the adversary corrupts some client $\mathcal{C}_j$, then each symmetric key $\mathrm{k}_{i,j}^{(0)}$ ($i \in [n]$ s.t. $\mathcal{C}_i \notin \mathbf{C}$) falls into the adversary's view, hence so does each $\mathbf{y}_i$. By Lemma 6.1 in [3], we have that $\mathcal{H}_0 \equiv \mathcal{H}_1$.

We define Sim by $\mathcal{H}_1$, and it follows that $\mathsf{Real}_{\Pi_0,P,C,\mathcal{A}}(1^\lambda, \{\mathbf{x}_i\}_{i\in[n]}) \equiv \mathcal{H}_0 \equiv \mathcal{H}_1 \equiv \mathsf{Ideal}_{\Pi_0,P,C,\mathcal{A},\mathsf{Sim}}\left(1^\lambda, \{\mathbf{x}_i\}_{i\notin T}, \mathbf{z}\right)$. $\qquad\square$

