# OpenReview forum: "Client-Private Secure Aggregation for Privacy-Preserving Federated Learning"
_NeurIPS.cc/2022/Workshop/Federated_Learning — FL-NeurIPS 2022 Poster_

### Official Review · Reviewer_o2FQ · 2022-10-17
**Comments**

This paper focuses on protecting both clients’ updates and the global model to achieve privacy-preserving federated learning. The authors propose a three-round secure aggregation protocol to encrypt clients’ updates and the averaged global model, which can only be decrypted by the clients. Finally, this work compares the overhead and runtime with prior works.

## Pros

- This work considers the privacy leakage of both clients’ updates and the global model.
- This paper provides the theoretical evidence for the correctness and security of the proposed protocol as well as empirical evaluation.

## Cons

- The improvement of protocol $\prod_A$ over protocol $\prod_B$ is not obvious. From the appendix, they all satisfy the same security level (line 519 and line 571), while the running time of \prod_A is noticeably higher than that of \prod_B (32x~8x for the different number of clients). Please discuss how to balance the trade-off.
- The security proof of $\prod_A$ needs further clarification. Adversaries with corrupted clients could directly obtain the protocol output. Why the S2 security is still satisfied for $\prod_A$?
- The effectiveness of DP needs more explanation. The authors illustrate the general process of DP which does not exist in the explicit construction of $\prod_A$. What impact will DP have on this protocol?
- More ablation studies on different dimensions(m) and modulus(q) should be provided.
- minor issues:
    line 181, compare its performance ``with'' $\prod_B$...

---

### Official Review · Reviewer_B6Y5 · 2022-10-17
**Federated Learning without the server seeing the model parameters**

This theory paper adapts Federated Learning with Secure Aggregation to a setting where the model parameters should not be visible to the server.

The paper is clear and coherent, and the algorithms presented seem to be solid.  The experimental results are appreciated, but it stands out strongly to this reviewer that the authors chose m (vector length) of only 10^2 elements -- this is dramatically smaller than most applications of federated learning in the very common deep learning setting, where 10^5-10^7 would be expected.

The limited experimental results highly a significant weakness in the presented algorithms: a per-client communication complexity of O(m*n), where n=number of clients.  This communication complexity would be prohibitive for cross-device federated learning even with reasonably small values of m and n; even for cross-silo learning, this would be significant as m and n grow.  Existing algorithms, such as the sited [3] have O(m+n) communication complexity -- a very significant distinction that is glossed over by the authors here.  It is also the case that the algorithms presented here quietly drop other practically important features compared to existing work, such as robustness to clients failing to complete the entire protocol.

Given the loss of features and the increased communication budget, once must ask whether the capabilities gained here could be obtained in any other way.  Unfortunately, the answer here seems to be yes -- that there are probably simpler methods than that authors' proposal.  All that is truly needed is for the clients to privately agree to a random vector before executing an existing secure aggregation protocol; by including this random vector among the inputs (e.g. by having one of the client devices add it to their own input), the modular sum the server learns through an existing protocol would still be blinded and completely opaque.  The blinded sum could be distributed to the clients, who could then each subtract off the the random blinding mask in order to obtain the correct value.   This strategy could easily be adapted to having more than one client add a blinding mask.  Sampled blinding masks could be distributed efficiently among clients as seeds to a PRNG.

Given the limitations of the proposed algorithm, and the fact that simpler modifications to existing algorithms seem to exist that do not require the same limitations, the overall impact of this work unfortunately seems rather limited to me.  I've rated the paper a 5, based on this limited anticipated impact; however, if the committee feels that a well-structured paper is more important that anticipated impact for this venue, I could support a rating of 6.

---

### Official Review · Reviewer_hhTM · 2022-10-18
**Review of Paper69**

This paper raised new aspect of secure aggregation protocol for federated learning (known as privacy-preserving federated learning), in which the server computes an encrypted global model that only clients can decrypt. That is, with the proposed client-private secure aggregation protocol, the server cannot learn the global model which includes information of training dataset of the clients while the server can learn the global model in conventional secure aggregation protocols. This paper theoretically proves correctness and security model, and empirically demonstrates its practicality.

While agreeing with that this paper raised the new aspect of secure aggregation protocol, the reviewer has the following major concerns.
1.	One of major concern is the contribution and novelty of this paper. The proposed algorithm consists of well-known building blocks, secret sharing and additive homomorphic encryption (AHE). O(mn^2) computational complexity is also not surprising, and even required to be further reduced as quadratic overhead of secure aggregation is main bottleneck to scale to millions of clients in cross-device FL.
2.	I admit that it raised the new perspective, but I am not sure about the necessity of this privacy model. It is easily broken when the server can collude with a single client.
3.	In the experimental results, the number of clients and dimension of model parameters are too small (up to 250 users and 100 features). To show the practicality and scalability of the proposed scheme, they are too small comparing with up to millions of clients and millions of model features in real-world FL settings.

---

### Decision · Program_Chairs · 2022-10-20

Accept (Poster)